# AttentionSmithy: A Modular Framework for Rapid Transformer Development

**Caleb Cranney**                                                    *caleb.cranney.app@gmail.com*
*Cedars Sinai Medical Center*
*Department of Computational Biomedicine*

**Jesse G. Meyer**                                                   *jessegmeyer@gmail.com*
*Cedars Sinai Medical Center*
*Department of Computational Biomedicine*

**Reviewed on OpenReview:**

## Abstract

Transformer architectures have revolutionized a broad spectrum of AI applications by leveraging attention mechanisms for parallelized and long-range sequence processing. Despite their remarkable success, building and customizing transformers remains prohibitively complex for many domain experts who lack deep knowledge of low-level implementations. We introduce AttentionSmithy, a modular software package that lowers the barrier to transformer innovation by decomposing key components—attention modules, feed-forward networks, normalization layers, and positional encodings—into reusable building blocks. By disentangling architectural elements into well-defined interfaces, users can rapidly prototype, adapt, and evaluate transformer variants without extensive coding overhead. Our framework currently supports four distinct positional encoding strategies (sinusoidal, learned, rotary, and ALiBi), offers modular integration of multiple attention methods (including standard attention, Longformer, and Linformer), and integrates seamlessly with neural architecture search (NAS) for automated design exploration. The system is designed to support future extensions with minimal overhead. We validate AttentionSmithy by replicating the original "Attention Is All You Need" transformer under resource constraints, demonstrating robust performance on a machine translation task. Leveraging the package's integrated NAS capability, we identified an optimized model configuration that outperformed our baseline, demonstrating the framework's effectiveness for automated architecture search and model improvement. We further illustrate AttentionSmithy's adaptability through gene-specific modeling, where a variant of a BERT-style architecture achieves over 95% accuracy on downstream cell type classification tasks using ranked transcriptomic data. These case studies underscore AttentionSmithy's core advantage: enabling specialized experimentation across diverse application domains—from natural language processing to genomic analysis—by obviating the need for labor-intensive, low-level framework manipulation. We anticipate that AttentionSmithy will serve as a foundation for creative transformer-based solutions, expediting research and development in numerous scientific and industrial fields.

## 1 Introduction

The transformer architecture (Vaswani et al., 2023) has revolutionized artificial intelligence, fundamentally changing how we approach sequence processing tasks across diverse domains. As transformer-based models continue to drive technological advancement and reshape societal interactions (Haque & Li, 2024), there is growing interest in adapting these architectures for specialized applications. However, customizing transformer architectures remains a significant challenge, requiring deep expertise in both the theoretical

foundations and implementation details. This complexity creates a barrier for domain experts who could otherwise leverage transformer capabilities for novel applications.

## 1.1 Transformer Architecture Fundamentals

While traditional recurrent neural networks like Long Short-Term Memory (LSTM) networks (Hochreiter & Schmidhuber, 1997) excelled at processing sequential data, they faced inherent limitations in parallelization and capturing long-range dependencies. The transformer architecture introduced by Vaswani et al. (2023) overcame these constraints through its innovative attention mechanism, enabling unprecedented advances in natural language processing (Patwardhan et al., 2023), computer vision (Pereira & Hussain, 2024), healthcare applications (Nerella et al., 2024), molecular science research (Jiang et al., 2024), and genomic analysis (Choi & Lee, 2023).

The basic building blocks of a transformer include [**Figure 1A**]:

1. Multi-head attention layers that compute and weigh relationships between all sequence elements in parallel

2. Feed-forward neural networks that process these relationships through non-linear transformations

3. Layer normalization components that stabilize training by normalizing activations across features

4. Residual connections that facilitate gradient flow and help preserve and reuse features from earlier layers

5. Positional encoding mechanisms that preserve sequence order information by encoding relative or absolute positions

## 1.2 Positional Encoding Strategies

A crucial aspect of transformer architectures is their handling of sequential information through positional encodings. Without such encodings, transformers would treat input sequences as unordered sets of tokens, losing critical information about both absolute positions (where exactly a token appears in the sequence) and relative positions (how tokens are ordered with respect to each other). For instance, the sentences "the dog chased the cat" and "the cat chased the dog" contain identical tokens but convey opposite meanings, while "chased cat dog the the" is syntactically invalid – distinctions a transformer could not make without position information. While the self-attention mechanism excels at capturing relationships between tokens, it is inherently permutation-invariant, necessitating an explicit method to encode positional context. Several strategies have emerged, each with unique implementation requirements:

Sinusoidal positional encodings (Vaswani et al., 2023) and learned positional embeddings (Wang & Chen, 2020) operate by adding position-specific vectors directly to input token embeddings. This straightforward approach allows for easy implementation but may have limitations in capturing relative positions effectively.

Rotary positional embeddings (Su et al., 2023) take a different approach, modifying the attention computation itself by applying rotation transformations to the query and key matrices. This method has shown particular promise in capturing relative positioning information while maintaining consistent attention patterns across sequence lengths.

ALiBi positional encodings (Press et al., 2022) introduce position-specific bias terms to the attention score matrix, effectively modulating the attention weights based on relative positions. This approach has demonstrated advantages in extrapolating to longer sequences than those seen during training.

## 1.3 Scalable Attention Strategies

Transformer models have set new benchmarks across a wide range of tasks, but their core attention mechanism scales poorly with sequence length, requiring quadratic time and memory resources with respect to

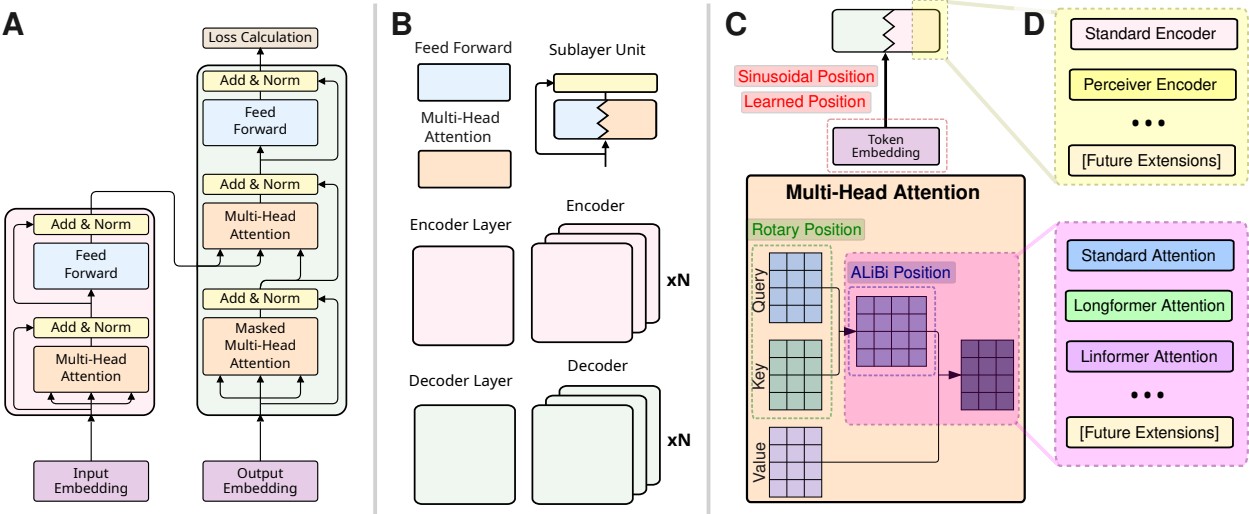

Figure 1: **Components of the transformer model architecture as coded in AttentionSmithy**. (A) The original transformer architecture introduced by Vaswani et al. (2023). (B) Labelled AttentionSmithy classes include implementations of the feed forward network, multi-head attention, sublayer unit (consisting of layer normalization and a residual connection surrounding an exchangeable feed forward network or multi-head attention class), encoder/decoder layers, and full encoders/decoders (which consist of an "N" number of layers determined by the end user). (C) Explicitly encoding position is a requirement of transformer models, but how position is encoded varies dramatically in implementation requirements from strategy to strategy. Four common strategies and their implementation details are outlined. The sinusoidal and learned positional embedding methods (red) involve directly adding vectors representing absolute positions to token embeddings before entering encoder or decoder layers. The rotary positional embedding method (green) requires adjusting the query and key matrices of the attention calculation directly. The ALiBi position embedding method (blue) adds negative values to the output of the query and key matrix multiplication that accumulate over greater positional distances. (D) The multi-head attention module accepts interchangeable attention methods, enabling support for strategies such as standard, Longformer, and Linformer attention. Additionally, base component classes can be swapped at the architectural level—for example, replacing a standard encoder with a Perceiver-style encoder. The modular structure also supports future extensions with minimal changes to surrounding components.

input size. This limitation restricts their applicability to long-context scenarios such as document processing, audio modeling, and genomic sequences. To enable practical use in these domains, various alternatives have been developed that reduce the computational cost of attention while extending its contextual capacity.

Some strategies adjust the attention calculation directly to avoid quadratic complexity. Examples include Longformer, which replaces dense attention with a combination of local windows and global attention tokens (Beltagy et al., 2020). Other strategies include Linformer, which introduces low-rank projections of the key and value matrices, effectively approximating attention computation with a reduced-rank representation (Wang et al., 2020). These methods reduce the number of token-to-token comparisons required, transforming attention from an all-to-all operation into a sparse or compressed one, which substantially lowers the cost from quadratic to linear scale with respect to sequence length.

Other strategies are architectural in nature, introducing intermediate representations that reduce the intensity of scaling with input length. One such example is Perceiver, which uses a small set of learnable latent vectors that attend to the input tokens (Jaegle et al., 2021). This replaces full self-attention over the input with a cross-attention mechanism, where computational cost scales with the product of the input length and the number of latent vectors—resulting in significantly lower complexity when the number of latents is much smaller than the input size.

## 1.4 Architectural Experimentation and Search

The modular nature of transformer architectures presents significant opportunities for systematic architectural exploration. Neural architecture search (NAS) has emerged as a promising approach for discovering optimal neural network configurations, but its application to transformers remains limited. While specialized NAS frameworks have been developed for transformers (Chitty-Venkata et al., 2022; Liu et al., 2022), they are typically purpose-built for specific research objectives, making it difficult for practitioners to adapt them for novel transformer architectures and unique application domains. Traditional implementations often tightly couple architectural elements, making it challenging to define a comprehensive search space that includes variations in attention mechanisms, positional encodings, and feed-forward networks.

## 1.5 Current Tools and Limitations

While widely used libraries like Hugging Face Transformers (Wolf et al., 2020), PyTorch (Ansel et al., 2024), and TensorFlow (Abadi et al., 2015) offer robust implementations of standard transformer architectures, they provide only limited flexibility for deep architectural customization. Core components are often tightly embedded within specific models, making it difficult to experiment with alternative encoding strategies, novel attention mechanisms, or entirely new architectural arrangements. This tight coupling not only hinders manual experimentation but also complicates the implementation of automated architecture search strategies, often forcing researchers to modify large sections of internal code or start anew.

Several specialized tools have emerged to improve transformer customization, each with valuable strengths. Modular Transformers (HuggingFace, 2025) stands out for its integration into the Hugging Face ecosystem, allowing users to easily swap components within pre-defined model templates. This streamlines workflows, particularly when one aims to modify or extend existing popular models. However, its focus on component swapping within fixed templates makes it challenging to restructure models at a deeper architectural level—such as building a Perceiver (Jaegle et al., 2021)—without significant additional work.

X-transformers (Wang, 2020) stands out as an impressive collection of experimental transformer features drawn from cutting-edge research, accessible through flexible parameters; however, because its core relies on dense conditional logic rather than modular building blocks, it becomes difficult to implement fundamental structural innovations—such as incorporating nonstandard patterns like Longformer attention (Beltagy et al., 2020) —without extensive code modification.

nanoGPT (Karpathy, 2022) takes a different approach, focusing on simplicity and clarity for those interested specifically in GPT models. Its compact PyTorch implementation has made it very popular for introductory experimentation, fine-tuning tasks, and tailoring to specific project needs. However, by design, it is specialized toward autoregressive GPT-style architectures, making it unsuitable for general-purpose transformer experimentation or architectures requiring encoder components or cross-modal innovations.

In this landscape, we present AttentionSmithy, a novel software package designed to democratize transformer development by providing a modular, component-based framework inspired by established software design principles (Ousterhout, 2018; Vogel et al., 2011; Gamma et al., 1995). Unlike existing tools that excel at customization within fixed boundaries, AttentionSmithy breaks transformers into flexible, reusable components, enabling rapid prototyping, systematic architecture search, and the creation of fundamentally new designs. Its modular architecture allows researchers to interchange and experiment with a wide range of features—including positional encoding strategies, attention mechanisms, and architectural configurations—tailoring models to their specific needs. By filling a crucial niche for maximal flexibility while maintaining architectural clarity, AttentionSmithy offers researchers and engineers a powerful toolkit for exploring and innovating beyond the limits of standard transformer design.

## 2 Methods

### 2.1 Software Architecture

Our software package implements a component-based design philosophy to facilitate the creation of customized transformer architectures. The core architecture breaks down transformer components into modular, reusable units that can be easily assembled and modified. This approach enables researchers to experiment with architectural variations while maintaining code readability and understanding of the underlying mechanisms.

The implementation utilizes PyTorch as its foundation and comprises distinct classes for each major transformer component. These components include multi-head attention mechanisms, feed-forward networks, normalization and dropout layers (implemented together as a "sublayer unit"), encoder/decoder layers, and complete encoder/decoder structures [**Figure 1B**]. Additionally, we provide both greedy and beam search generators for sequence generation tasks.

Appendix materials include a description for an example training script performing a machine translation task [**Appendix A.1**], as well as short code snippets illustrating how to build and customize transformer components using AttentionSmithy [**Appendix A.2**].

### 2.2 Key Features

#### 2.2.1 Flexible Positional Encoding Framework

A key architectural feature is the implementation of a positional embedding strategy pattern that manages various numeric embedding approaches. A strategy manager serves as an intermediary for selecting and applying different positional encoding implementations within the transformer architecture, allowing for seamless integration of different approaches without requiring modifications to the core architecture.

Our implementation currently supports four distinct positional encoding strategies: sinusoidal, learned, rotary, and ALiBi embeddings, chosen as representative examples of popular approaches in the field. Each strategy is implemented as a separate class and managed through the embedding strategy manager, with the architecture designed to readily accommodate additional encoding strategies as they emerge. This flexibility is particularly valuable because different positional encoding strategies require fundamentally different implementations within the transformer architecture: sinusoidal and learned positional embeddings are added to input token vectors, rotary positional embedding requires adjusting the query and key matrices in the attention calculation, and ALiBi adds values to the attention score matrix (the product of the query and key matrices) [**Figure 1C**]. Our extensible design allows users to activate or deactivate these varied encoding strategies independently, enabling direct comparisons of their effectiveness in various applications, while also providing a framework for implementing and testing novel positional encoding approaches.

Beyond traditional position representation, the modular implementation of positional encodings in AttentionSmithy enables these methods to be applied to any numeric data type. This opens new possibilities for representing temporal, quantitative, or other ordered information within transformer architectures. For instance, in time-series analysis, researchers could simultaneously encode both sequential position and temporal features using different encoding strategies.

#### 2.2.2 Modular Attention Mechanisms

The system implements a modular multi-head attention framework that supports interchangeable attention variants, specified at initialization. This design accommodates a range of scalable attention mechanisms without modifying the core architecture. The framework currently includes implementations of sparse attention (e.g., Longformer), low-rank approximation (e.g., Linformer), and architectural adaptations (e.g., Perceiver) [**Figure 1D**]. This modular design enables rapid prototyping and integration of new attention strategies with minimal architectural disruption. Notably, while current implementations of Longformer, Linformer, and Perceiver attention are applied only within encoder architectures, the modular design readily supports adapting to decoder and encoder-decoder settings for strategies appropriate for those architectures.

### 2.2.3 Neural Architecture Search Compatibility

The modular design of AttentionSmithy facilitates automated architecture optimization through neural architecture search (NAS). Components can be easily swapped or modified programmatically, allowing for systematic exploration of architectural variations while maintaining code interpretability.

The NAS workflow was based on the "Multi-Objective NAS with Ax" workflow tutorial on the official PyTorch website, utilizing Meta's Ax package to do so (Eriksson & Balandat, 2022; Meta, 2025). The process includes designing a search space as a separate Python script that accepts variables dictating the model structure, setting up a TorchX runner and scheduler for submitting model training scripts concurrently, and defining optimization requirement configurations. Ax uses Bayesian optimization to evaluate and compare model configurations and their predictive accuracy, highlighting the impact specific architectural decisions have on the final loss.

For the machine translation task, we used the BLEU score (Papineni et al., 2002) (reported on a scale of 0–100) as the primary evaluation metric, consistent with the original transformer paper (Vaswani et al., 2023). The search space consisted of six adjustable parameters: each of the four implemented positional encoding methods could be toggled on or off, the dropout rate, and the activation function used in the feed-forward network. Models were trained for five epochs during the search to reduce time complexity.

To demonstrate how domain experts can apply NAS to specialized applications using AttentionSmithy, we extended the NAS workflow to the Geneformer task. The search space included three positional encoding strategies (sinusoidal, learned, and rotary), dropout rate, activation function, and attention mechanism. Additionally, nonstandard attention types included task-specific hyperparameters: Longformer (context window length), Linformer (projected key dimension), and Perceiver (number of latent encoder layers and latent space length). Each trial was run for six thousand steps with a batch size of 32.

### 2.3 Code Availability

The source code for AttentionSmithy is publicly available on GitHub (https://github.com/xomicsdatascience/AttentionSmithy). The code implementing machine translation is also available at https://github.com/xomicsdatascience/machine-translation and utilizes the WMT14 German-English dataset (Bojar et al., 2014) accessed through the Hugging Face datasets library. The code implementing geneformer (Theodoris et al., 2023) is available at https://github.com/xomicsdatascience/geneformer, utilizing preprocessed data from the original geneformer implementation HuggingFace repository. All code repositories are released under the MIT license. The software originated from a re-implementation of code depicted in the Annotated Transformer article (Rush et al., 2022).

AttentionSmithy is implemented in Python using PyTorch (Ansel et al., 2024). To enhance usability and standardization, AttentionSmithy is designed to be compatible with PyTorch Lightning (Falcon, 2019), allowing researchers to easily incorporate training loops, distributed training, and other advanced features while maintaining clean, research-focused code.

### 2.4 LLM assistance

Claude 3.5 Sonnet (Anthropic, 2024) and ChatGPT-4o (OpenAI, 2024) were used to help with writing this manuscript.

## 3 Results

### 3.1 Validation Studies

We conducted three validation studies to demonstrate the efficacy and versatility of AttentionSmithy: a replication of the original vanilla transformer model, an optimized model determined by a neural architecture search (NAS), and a bioinformatics application.

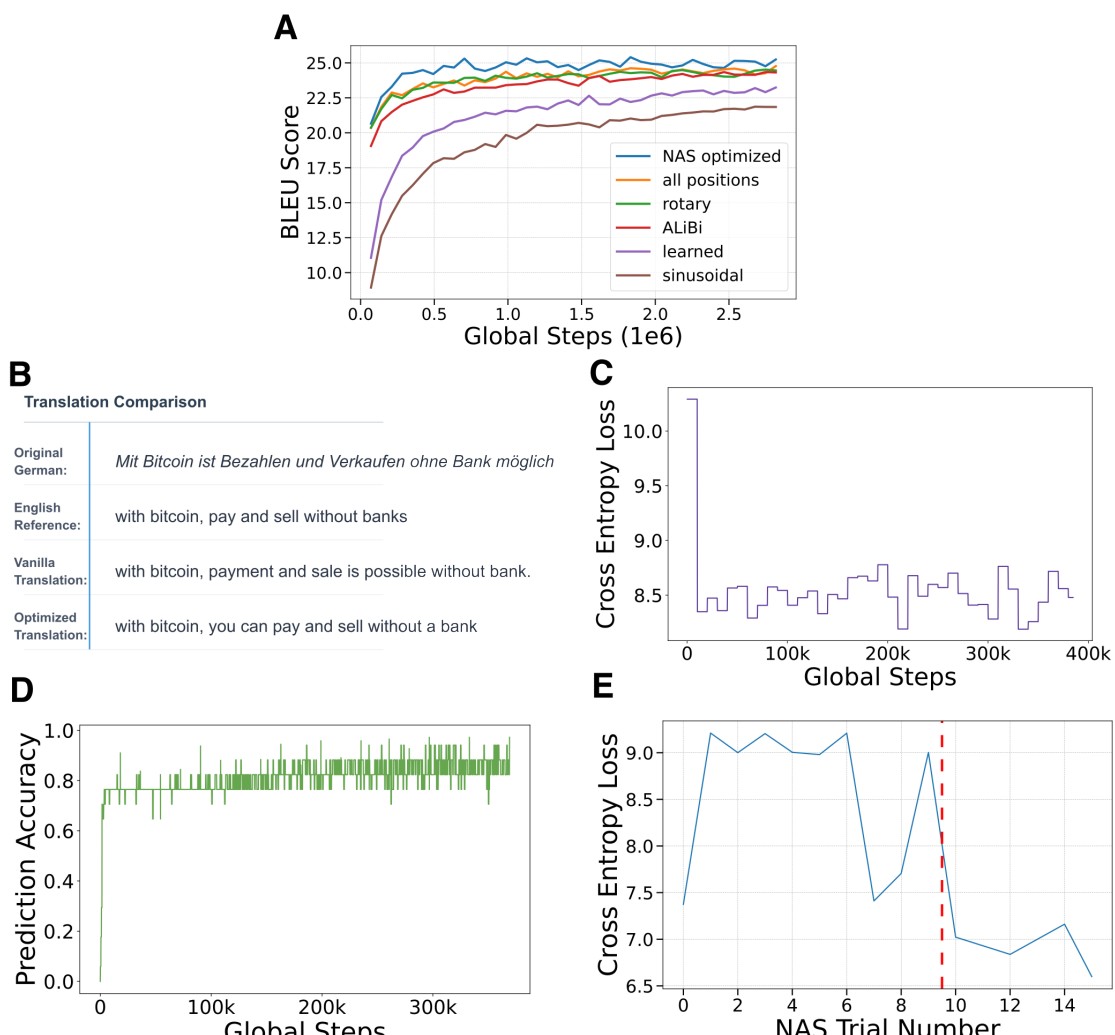

Figure 2: **Validation studies of programs built with AttentionSmithy.** (A) BLEU scores comparing translation quality across six transformer variants. The initial NAS-optimized model used a tanh activation function, a dropout rate of 0.0, and a combination of all positional encoding strategies (blue). For comparison, we evaluated models using the original transformer's settings—ReLU activation and 0.1 dropout—with different positional encodings: rotary (green), ALiBi (red), learned (purple), sinusoidal (brown), and the combined set of all four (orange). Note that the sinusoidal configuration corresponds to the original transformer model (Vaswani et al., 2023). (B) Representative examples of German-to-English translation outputs, showing source text, reference translation, and outputs from both NAS optimized and original transformer variants. The BLEU score in this specific example improved from 16.1 with the vanilla transformer to 34.4 with the optimized architecture. (C) Validation loss trajectory during pretraining of the Geneformer foundation model, plotted against global training steps. (D) Cell type classification accuracy on the validation set during fine-tuning of the pretrained Geneformer model, also plotted against global training steps. (E) Validation losses from 14 unique neural architecture search (NAS) trials targeting optimization of the Geneformer model. These trials represent a subset of 50 total runs, with duplicates removed. Trials 0–9, shown to the left of the red line, were pseudo-random configurations, while remaining trials were targeted explorations guided by NAS.

### 3.1.1 Original Transformer Replication

We implemented the transformer architecture and training setup described in Vaswani et al. (2023), using the WMT 2014 English-German dataset, which consists of 4.51M sentence pairs (approximately 9.03M sentences

total). Our primary training run was limited to a maximum context window of 100 tokens, mainly to reduce training time given the use of a single A100 GPU. While this constraint was partly driven by computational efficiency, it was also informed by the nature of the dataset: only 50,860 sentence pairs (approximately 1.1%) included at least one sentence longer than 100 tokens and were excluded from training. To confirm that this truncation had minimal effect on overall performance, we conducted an additional run using a 500-token context window (see **Appendix A.3**). This expanded setting excluded only 398 sentence pairs (less than 0.01%) with at least one sentence exceeding 500 tokens, and was comparable to the same model with a 100-context window constraint. After 40 epochs of training, our 100-token model achieved a BLEU score of approximately 21 on the machine translation task (**Figure 2A**, brown line). While this falls short of the original paper's BLEU score of 25, it represents a reasonable outcome given the resource constraints.

To highlight the ease of NAS enabled with AttentionSmithy, we designed a search space around the original model components to identify architectural changes that may enhance performance. The optimized model from NAS had a more rapidly increasing BLEU score across training steps, and in the end achieved a higher BLEU score of approximately 24 after 40 epochs [**Figure 2A**, blue line], approaching the performance of the original implementation despite our hardware constraints. Key modifications identified from NAS included: simultaneous utilization of all four positional encoding strategies (sinusoidal, learned, rotary, and ALiBi), removal of dropout (reduction from 0.1 to 0.0), and replacement of ReLU with tanh activation in feed-forward networks. This led to generally better translations, an example of which is shown in **Figure 2B**.

An initially intriguing outcome from our neural architecture search was the apparent success of configurations using all four available positional encoding methods simultaneously. However, follow-up ablation experiments clarified that this combination did not yield additive or synergistic benefits; rather, the improvements were likely driven primarily by the use of rotary positional encoding, which performed approximately as effectively on its own as the combined setup [**Figure 2A**]. This underscores the importance of validating NAS-derived architectures through targeted ablation experiments to identify the true sources of performance gains.

### 3.1.2 Bioinformatics Application

To demonstrate domain adaptability, we replicated the Geneformer model for transfer learning for transcriptomic single-cell data tasks (Theodoris et al., 2023). Following the original paper's methodology, we pre-trained the model using a BERT-style architecture on rank-based transcript data, with genes serving as tokens. Importantly, to ensure the model had sufficient capacity to learn long-range dependencies, we used a maximum context window of 2048 tokens—matching the configuration in the original Geneformer implementation. To verify that the model captures contextual information even after plateauing [**Figure 2C**], we fine-tuned it for cell type classification using this dataset, freezing the first two pre-trained layers and adding a classification layer as specified in their methodology. We used their published human_dcm_hcm_nf dataset for this task, which contains 579,159 cells representing 21 distinct cell types from cardiac tissue from 29 individuals. This implementation achieved over 95% accuracy on the validation dataset, demonstrating successful replication of the Geneformer architecture's ability to transfer contextual relationship information for downstream gene expression analyses [**Figure 2D**].

As a proof of concept, we additionally applied NAS to the Geneformer task to explore potential improvements. The NAS identified a configuration combining Longformer attention with a local attention window of 32, no positional encoding, zero dropout, and leaky ReLU activation, which achieved a validation loss of 6.60 [**Figure 2E**]. Specialized linear projections for global attention in Longformer can be enabled in AttentionSmithy, though this option was not included in the Geneformer NAS task, and thus not included in the optimal model. While we did not pretrain or fine-tune this optimized architecture in the current study, the reduction in validation loss relative to the baseline model (which plateaued at 8.0–8.5, as seen in **Figure 2C**) suggests promising avenues for future exploration.

## 4  Discussion

While our validation studies focused on established architectures, they serve primarily to demonstrate AttentionSmithy's foundational reliability. The package's true value lies in enabling researchers to develop entirely

new transformer architectures for specialized applications that may not yet exist. By providing a flexible, modular framework, we empower domain experts to experiment with novel combinations of transformer components without requiring deep expertise in transformer implementation details.

This capability is particularly valuable in scientific domains where traditional transformer architectures may not perfectly fit the underlying data structures or research questions. For instance, researchers working with complex multimodal data could leverage our framework to develop hybrid architectures that process different data types through specialized attention mechanisms. The ability to experiment with multiple positional encoding strategies simultaneously opens new possibilities for representing complex relationships in data, whether they be spatial, temporal, or domain-specific ordered relationships.

The modular nature of AttentionSmithy enables researchers to focus on the unique aspects of their application domains rather than becoming entangled in transformer implementation details. This democratization of transformer development has the potential to accelerate innovation in fields where artificial intelligence applications are still emerging. For example, researchers could apply self-supervised speech representation techniques (Mohamed et al., 2022) to nanopore sequencing, enabling efficient and accurate nucleotide sequencing through pre-training and fine-tuning approaches. In mass spectrometry, developing foundation models to interpret data-independent acquisition (DIA) spectra could allow researchers to leverage these complex, chimeric signals for downstream tasks without relying on pre-existing spectral libraries. The transformative potential of this architecture extends well beyond current applications, and we anticipate that researchers across diverse scientific domains will develop innovative implementations that we cannot yet foresee.

Future development of AttentionSmithy will focus on expanding its capabilities to support emerging transformer variants while maintaining its commitment to architectural clarity and ease of use. We encourage contributions from the research community, particularly in implementing new positional encoding strategies and exploring applications in specialized domains. This could include relative positional embeddings (Shaw et al., 2018) and their T5 variant (Raffel et al., 2023), which focus explicitly on the relationships between positions rather than absolute positions. Through continued development and collaboration, we aim to further lower the barriers to entry for transformer architecture experimentation and innovation across scientific disciplines.

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

## A Appendix

### A.1 Supplementary Code

The included supplementary `.zip` file (`machine-translation.zip`) contains example code demonstrating the implementation and execution of a machine translation transformer model using AttentionSmithy. The included scripts facilitate data downloading, model training, and evaluation.

#### A.1.1 Contents

- `data_download.py` – Downloads and preprocesses the WMT-14 German-English dataset.
- `data_import.py` – Handles dataset loading and processing for training.
- `main.py` – Runs the model training and evaluation pipeline.
- `model_import.py` – Defines the translation model and its components.
- `README.md` – Provides setup instructions, expected outputs, and additional notes.

### A.2 Short Code Examples

The AttentionSmithy package provides modular components for constructing custom transformer architectures. It supports encoders, decoders (not shown in this sample), and full encoder-decoder stacks, with swappable attention mechanisms and experimental numeric embedding strategies.

#### A.2.1 Assembling a Transformer Encoder

We start by building an encoder from modular pieces:

- `MultiheadAttention` (attention mechanism)
- `FeedForwardNetwork` (nonlinear transformation)
- `EncoderLayer` (single transformer block)
- `Encoder` (stacked layers)

While this sample focuses on encoders, decoder layers are fully supported and can be constructed analogously.

Listing 1: Basic Encoder Assembly

```
1 from attention_smithy.components import Encoder, EncoderLayer,
    MultiheadAttention, FeedForwardNetwork
2 from attention_smithy.attention import StandardAttentionMethod
3
4 embedding_dim = 512
```

```
5  num_heads = 8
6  feedforward_dim = 2048
7  dropout = 0.1
8
9  # Define standard multihead attention
10 attention = MultiheadAttention (
11     embedding_dimension = embedding_dim ,
12     number_of_heads = num_heads ,
13     attention_method = StandardAttentionMethod ( dropout = dropout )
14 )
15
16 # Define feedforward block
17 feedforward = FeedForwardNetwork (
18     embedding_dim , feedforward_dim , activation = 'relu ', dropout = dropout
19 )
20
21 # Wrap in encoder layer
22 encoder_layer = EncoderLayer (
23     embedding_dimension = embedding_dim ,
24     self_attention = attention ,
25     feed_forward = feedforward ,
26     dropout = dropout
27 )
28
29 # Stack into multi - layer encoder
30 encoder = Encoder ( encoder_layer , number_of_layers =6)
```

### A.2.2 Using Alternative Attention Mechanisms

AttentionSmithy supports experimental attention patterns, such as Longformer's local-global windowing or Linformer's projection-based sparsity. These can be swapped seamlessly with standard attention via the `attention_method` argument of the `MultiheadAttention` class.

Listing 2: Using Longformer Attention

```
1  from attention_smithy.attention import LongformerAttentionMethod
2
3  # Use Longformer - style local+global attention
4  longformer_attention = MultiheadAttention (
5      embedding_dimension = embedding_dim ,
6      number_of_heads = num_heads ,
7      attention_method = LongformerAttentionMethod (
8          attention_window =256 ,
9          dropout = dropout
10     )
11 )
```

### A.2.3 Adding Positional and Numeric Embeddings

The `NumericEmbeddingManager` allows combining multiple numeric embedding strategies, applied at different pipeline stages:

- at the token embedding step (e.g., Sinusoidal, Learned)

- before attention score calculation (e.g., Rotary)

- after attention score calculation (e.g., ALiBi)

Developers can easily implement new positional embedding strategies and provide them at numeric embedding manager initialization, making it straightforward to experiment with alternatives like sinusoidal, learned, rotary, or ALiBi embeddings. Additionally, the design supports strategies for encoding **non-positional numeric features**, such as time intervals, distances, or other custom scalar inputs, further increasing the model's flexibility and representational capacity.

Listing 3: Numeric Embedding Manager

```python
from attention_smithy.numeric_embeddings import (
    SinusoidalPositionEmbedding, LearnedPositionEmbedding,
    RotaryPositionEmbedding, ALiBiPositionEmbedding,
    NumericEmbeddingManager
)

# Initialize multiple strategies
embedding_manager = NumericEmbeddingManager([
    SinusoidalPositionEmbedding(embedding_dim),
    LearnedPositionEmbedding(max_sequence_length=3000,
        embedding_dim=embedding_dim),
    RotaryPositionEmbedding(rotary_dimension=embedding_dim // num_heads),
    ALiBiPositionEmbedding(num_heads=num_heads)
])

# Generate combined sinusoidal and learned positional embedding
token_embedding = some_input_embedding_tensor
positional_embedding = embedding_manager.create_positional_or_custom_embedding(
    token_embedding=token_embedding
)

# Add positional embedding to token embedding
final_embedding = token_embedding + positional_embedding
```

**A.3   Supplementary Figure 1**

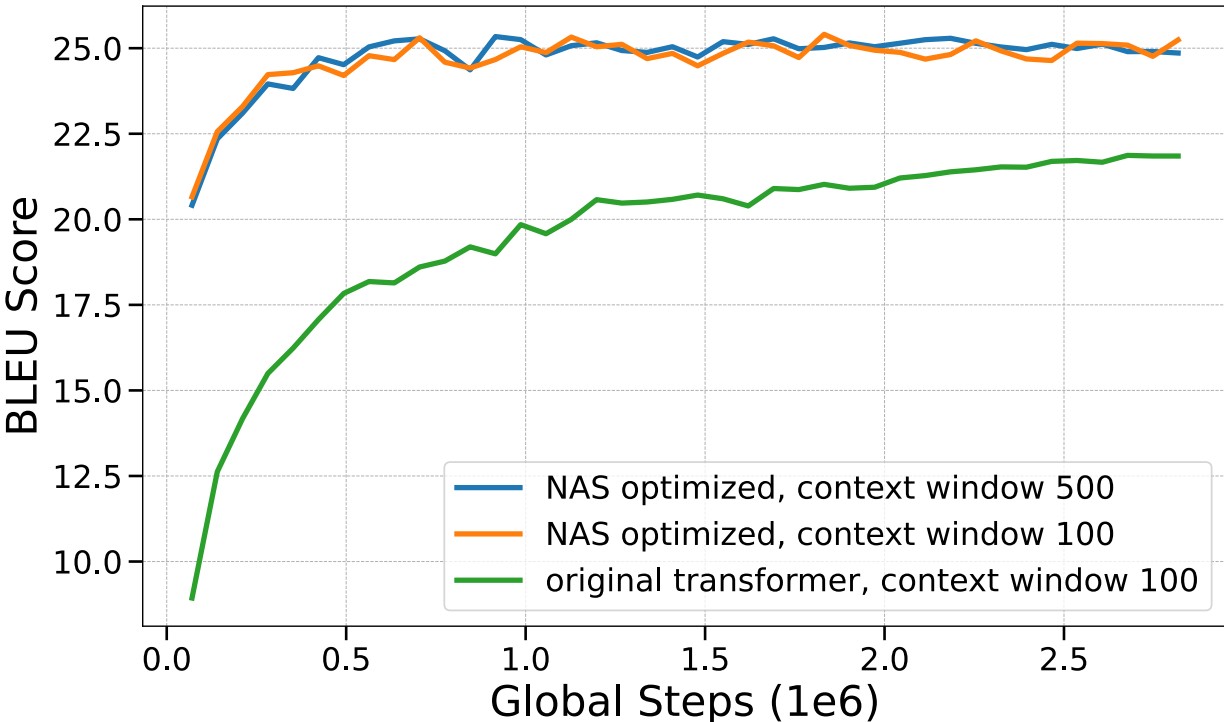

Supplementary Figure 1: **BLEU score evaluation for differing context window lengths.** NAS-optimized transformer with 500-token context window (blue), NAS-optimized transformer with 100-token context window (orange), and the original transformer baseline with 100-token context window (green). The NAS-optimized models substantially outperform the baseline, with minimal difference between the 500- and 100-token context window settings.

