# OpenReview forum: "AttentionSmithy: A Modular Framework for Rapid Transformer Development"
_TMLR — Accepted by TMLR_

### Review · Reviewer_L4jA · 2025-04-13

**Summary Of Contributions:**

This paper presents AttentionSmithy, a modular software package that ease the pre-training of custom transformer-like models by packing transformer modules into building blocks. This package also incorporate a small search space containing different positional encoding methods, dropout rate and activation functions, where users can use built in NAS algorithm to gain potential performance boost.

**Audience:**

Yes

**Claims And Evidence:**

Yes

**Requested Changes:**

Please address the points above to strengthen the claims in section 4.

**Strengths And Weaknesses:**

Strengths:
1. The paper is well written and easy to follow.
2. The idea of this work is indeed intriguing.
3. The framework is evaluated on two different application domains, and the customized model found by NAS algorithm observes decent improvement compare to the baseline run.

Weaknesses:
1. No NAS result shown for the Bioinformatics application, currently it looks like the author just reproduce the original paper with their own framework. Is further improvement possible after NAS algorithm? Or simply adding the other encoding strategies can cause performance gain?
2. The results from NAS algorithm on machine translation task shows large improvement. However, there are multiple modifications identified from NAS, which does not support the conclusion made in section 4.1. An ablation on these individual modifications will help a a lot.
3. Given that the point of this package is to accelerate the development process, a comparison of effort required to build some custom transformer model using Huggingface for example and the proposed framework will be great (e.g. How many lines of code required?).

---

> ### Author Response · Authors · 2025-04-15
> **Response to Reviewer L4jA**
>
> We sincerely thank you for your thoughtful and constructive review. We greatly appreciate your positive comments on the clarity of our writing, the conceptual utility of AttentionSmithy, and the empirical results across two application domains. Below, we address each of your concerns and outline the changes we plan to make in the revised manuscript.
>
> ---
>
> ### 1. No NAS result shown for the bioinformatics application
> You raise an important point. In the current version, the Geneformer-based bioinformatics task was reproduced without modification, and we agree this does not fully demonstrate the power of AttentionSmithy to explore architectural variants.
> - To address this, we will create a NAS-optimized version of the Geneformer model and compare its fine-tuning performance to the original.
>
> ---
>
> ### 2. Lack of ablation analysis for NAS-optimized machine translation model
> Thank you for highlighting this limitation. While our NAS results suggest that combining multiple positional encoding strategies contributes to improved performance, we did not isolate their individual contributions in the original manuscript.
> - To address this, we will perform an ablation study in which we train two additional models:
>   - One using all four positional encodings,
>   - One using only two positional encodings.
> - All other variables will match the baseline transformer. These results will help determine whether the improvements are due to the stacked encodings and provide clearer support for the conclusions in Section 4.1.
>
> ---
>
> ### 3. Comparison to HuggingFace and development effort reduction
> We appreciate your suggestion to more clearly demonstrate the reduction in development effort enabled by AttentionSmithy.
> - In the revised manuscript, we will quantify the number of lines of code abstracted away through the use of AttentionSmithy class methods. Specifically, for the two examples presented in this paper, we will count and report the number of lines in the internal AttentionSmithy classes that were leveraged to produce the final models.
> - We will also include a supplement comparing the concise user-side code (e.g., the configuration and invocation scripts) to the full class code that would otherwise need to be written manually. This will help illustrate the modularity and code-saving benefits of our framework.
> - Additionally, in the **Discussion**, we will emphasize that many of the model architectures enabled by AttentionSmithy — such as those with mixed positional encodings and novel attention mechanisms — are not supported by existing libraries like HuggingFace, and would require significant manual engineering effort to implement from scratch.
> - We will also use the **Discussion** to underscore how AttentionSmithy is designed to pair naturally with architecture search, facilitating exploration of novel model variants that would otherwise be too time-intensive to prototype.
>
> ---
>
> Please let us know if there are additional improvements you would recommend. We again thank you for your thoughtful review and helpful suggestions.

---

> > ### Author Response · Authors · 2025-05-08
> > **May 8 Update**
> >
> > We are currently running the Geneformer NAS experiments, which will be included in the forthcoming revision. We have completed an ablation analysis on the machine translation task and updated the manuscript to reflect that, contrary to initial expectations, combining multiple positional encodings did not yield additive benefits past the optimal rotary method. Additionally, we expanded the comparison to other frameworks and included supplemental code snippets with line counts to highlight the reduction in development effort enabled by AttentionSmithy. The full revised manuscript, including the Geneformer NAS results, will be submitted by EOD May 14.

---

> > > ### Comment · Reviewer_L4jA · 2025-05-22
> > > **Reviewer comment**
> > >
> > > Thank you for your revisions and the additional experiments addressing my concerns. I would like to recommened acceptance.

---

### Review · Reviewer_5n9H · 2025-04-26

**Summary Of Contributions:**

This paper proposes AttentionSmithy, a modular framework designed to simplify and accelerate the customization of Transformer architectures. It breaks down key components such as attention mechanisms, feed-forward layers, and positional encoding into reusable building blocks. Its main contributions include a flexible system for experimenting with positional encodings (sinusoidal, learned, rotary, and ALiBi) and compatibility with neural architecture search (NAS) to optimize model configurations. Experiments confirmed its effectiveness by successfully replicating the original Transformer model and demonstrating improved performance through automated design exploration. Additionally, AttentionSmithy extended its applicability beyond NLP, achieving over 95% accuracy in genomic cell-type classification, showcasing its adaptability across scientific domains.

**Audience:**

Yes

**Broader Impact Concerns:**

None.

**Claims And Evidence:**

No

**Requested Changes:**

- Detail the advantages over current frameworks: In Section 1.4, the paper briefly mentions some limitations and states that “implementing customized architectures remains a complex undertaking that often requires building from scratch.” However, there are several existing options for easily building and testing novel transformer variants, namely:

  - Modular Transformers [1]: allows reusing components from existing models.

  - x-transformers [2]: provides several transformer components, including attention mechanisms and positional encoding options, which can be easily toggled.

  - nanoGPT [3]: a simple transformer implementation that is easy to modify and use in small-scale experiments.

What specific advantages does AttentionSmithy offer over the options above? Addressing this question could help improve the paper.

- If possible, provide experiments using longer sequences: There are small-scale experiments based on FineWeb [4] that could help demonstrate the flexibility of AttentionSmithy and validate the claims regarding the effectiveness of combining multiple positional encoding methods.

- Minor:

  - Provide links in the bibliography or footnotes for the “Multi-Objective NAS with Ax” workflow tutorial (page 4).

  - Provide links in the bibliography or footnotes for Claude 3.5 Sonnet and ChatGPT o1 (page 5).

  - Section 1.4, page 2: the correct framework name is Hugging Face Transformers.

- [1] https://huggingface.co/docs/transformers/en/modular_transformers
- [2] https://github.com/lucidrains/x-transformers
- [3] https://github.com/karpathy/nanoGPT
- [4] https://github.com/KellerJordan/modded-nanogpt

**Strengths And Weaknesses:**

### Strengths

- The paper presents a framework that arguably reduces the technical barriers to exploring Transformer architecture variants.
- Empirical evidence of potential applications is provided in both natural language and biological domains.

### Weaknesses

- The comparison with existing frameworks and tools is not sufficiently explored. Other tools are available for experimenting with transformers, and a more detailed comparison of their limitations could make the value proposition of this framework more evident. Refer to "Requested Changes" for more details.

- The experiments are limited to a very specific setting using short sequences, which weakens the claim that combining multiple positional encoding methods helps in practical problems.

- Besides the positional encoding variants, the framework still has a limited set of components for other important parts of the architecture, such as the attention mechanism.

---

> ### Author Response · Authors · 2025-05-01
> **Proposed changes**
>
> We are deeply grateful for your detailed and constructive feedback.
> 1.	Comparing AttentionSmithy to Modular Transformers, x-transformers, and nanoGPT
> Thank you for detailing specific programs you would like evaluated against AttentionSmithy. We will extend section 1.4 (Current Tools and Limitations) with a more thorough comparison that includes these repository code bases. Even though the most comprehensive alternative, x-transformers, is extensive, our package has several advantages that we will detail.
> 2.	Provide experiments with longer sequences
> As noted in response to reviewer teX3 comment #2, we added new attention methods that scale better to long sequences, and we plan to apply them in the geneformer task, and if time permits, to the geneformer NAS. We will also run a machine translation task with >100 token context window using the optimal model.
> 3.	Update bibliography and naming conventions
> Thank you for pointing out our bibliography and naming errors. We will update them in our next revision.
>
> Please let us know if these changes would be suitable.

---

> > ### Comment · Reviewer_5n9H · 2025-05-05
> > **Reviewer comment**
> >
> > Thanks for your reply. I believe the proposed modifications would address the requested changes.

---

> ### Author Response · Authors · 2025-05-08
> **May 8 Update**
>
> We have added several paragraphs to section 1.4 comparing AttentionSmithy to Modular Transformers, x-transformers, and nanoGPT, highlighting key differences and advantages. We are currently running the 500-token context window machine translation experiment, but so far we are not seeing a substantial impact. This is likely a limitation of the WMT14 German-English dataset – of the 4,508,785 English–German sentence pairs in the training dataset, only \~50,860 extra pairs have a sentence between 100 tokens and 500 tokens (\~1.1%) and just ~398 beyond 500, making it suboptimal for evaluating longer-context benefits. In addressing your request, we realized that we had not explicitly reported the maximum sequence length (2,048 tokens) used in the Geneformer experiments in the original draft—this was an oversight on our part, and we will make sure to include it in the revision. We have also updated the bibliography, including author attributions such as “OpenAI” where appropriate. The complete revision will be submitted by EOD May 14 as we await final analyses to complete.

---

> > ### Comment · Reviewer_5n9H · 2025-05-21
> > **Reviewer comment**
> >
> > I would like to thank the authors for the revisions, which strengthen the evidence supporting their claims and the utility of the proposed tool. I will take these changes into account in my final recommendation.
> >
> > Minor comment: The font sizes in Supplementary Figure 1 appear somewhat disproportionate relative to the main text.

---

### Review · Reviewer_teX3 · 2025-04-26

**Summary Of Contributions:**

The paper introduces AttentionSmithy, a modular framework for developing and customising transformer architectures, especially focusing on being accessible for domain experts without in-depth AI knowledge. It allows users to experiment with different parts of the transformer model, like attention modules, feed-forward networks, and positional encodings through a collection components that can be combined and configured. It is also designed to integrate well with neural architecture search (NAS) to enable automatic optimisation of transformer architectures. The current version supports various positional encoding strategies, and the framework is validated through experiments in machine translation and gene-specific modelling, demonstrating its effectiveness and adaptability across different domains.

**Audience:**

Yes

**Broader Impact Concerns:**

None needed.

**Claims And Evidence:**

No

**Requested Changes:**

**Critical changes:**

Revise the claim about achieving near state-of-the-art performance in machine translation, and expand the variety of attention mechanisms to fulfil the framework's promise as outlined in the detailed review.

**Other:**

See above section on minor concerns.

**Strengths And Weaknesses:**

**Strengths:**
* *Modularity:* The framework's modular design allows for easy experimentation and rapid prototyping, which is well-aligned with the current needs for agile development in AI research.
* *Neural Architecture Search Integration:* The integration of NAS makes it easier for users to automatically find optimal architectures for their domain.
* *Insights on positional encodings:* One of the experiments highlights the complementary benefits from using a combination of encoding methods.

**Weaknesses:**

My main concerns relate to the paper’s claims:
* *Demonstrating near state-of-the-art performance on a machine translation task.* This claim is too strong. The results seem to be competitive with the original Transformer paper from 2017, with possible signs of improvement when considering the compute constraints. But the current stat-of-the-art on the WMT German-English seems to be around a BLEU score of 35. This claim needs adjusting.
* *Attention Smithy’s core advantage: enabling specialized experimentation across diverse application domains—from natural language processing to genomic analysis—by obviating the need for labor-intensive, low-level framework manipulation.* In order for the framework to be truly useful for domain experts (i.e. non-AI-experts), it needs to offer more implemented components to search across. Especially with a name like AttentionSmithy., the most important next step is to offer more attention methods, as is discussed and shown in Supplementary Figure 1.

To recommend acceptance, these two main concerns need to be addressed. Below are a few minor concerns that could further improve the paper.

Minor concerns:
* I think the Geneformer experiment could be more meaningful. I assume the point here is to show that the framework works for non-standard domains. Ideally this experiment would also include a search, to show the benefits of this type of modular framework across different domains.
* Since the current paper is under the page limit, there would be space to add some short code example of either the positional encoding implementations or more importantly, a few lines of code showing how the framework is used.
* While it is somewhat orthogonal to the main contribution of the paper, it would be great to understand more about the complementary benefits from the different encoding mechanisms from the NAS results. Does the package provide any useful information for interpreting these types of results?
* What do the colours mean in the different attention methods in Supp Fig 1? The caption could give some info on this.

---

> ### Author Response · Authors · 2025-05-01
> **Proposed changes**
>
> We thank the reviewer for their thoughtful feedback.
> 1.	State-of-the-art performance claim correction: We apologize for this error. We will remove any claims that our performance is state-of-the-art and instead focus on the capabilities enabled by our package.
> 2.	Offer more attention alternatives: We added Longformer and Linformer attention methods. We will demonstrate their application on the Geneformer task. We regret that we don’t have enough time in the 4 week maximum review period to add additional attention methods, but we hope to add Big Bird and Performer as part of our development roadmap.
>
> We have also implemented a variant inspired by the Perceiver architecture, where attention is applied between a small set of learnable latent vectors and the full input sequence. Although Perceiver does not alter the core attention operation itself, it manipulates the attention pattern, shifting from full self-attention among inputs to a latent-to-input cross-attention scheme, representing a meaningful and impactful architectural variation. This addition further demonstrates the flexibility of AttentionSmithy to support diverse attention paradigms with minimal engineering overhead.
>
> If time permits, we plan to include these variants (i.e. Longformer, Linformer, and Perceiver) as options in a NAS evaluation for the Geneformer task, where their utility will be highlighted by enabling the models to access to the full gene token sequence without the need for windowing or truncation.
>
> Minor concerns:
> 3.	Perform a Geneformer NAS evaluation: Reviewer L4jA has expressed similar concerns that a neural architecture search was not performed with Geneformer, and we plan to implement such a search as part of our response (see above).
> 4.	Code examples: We had provided an example machine translation implementation in the supplement, and we agree we can make this clearer by adding a short code example to the paper as a supplemental figure (see also response to L4jA’s point #3).
> 5.	Understanding the complementary benefits of different positional encoding mechanisms:  also in response to reviewer L4jA, we plan on performing an ablation analysis comparing various positional encoding strategies (separate or combined)
> 6.	We will add a color legend to supplement figure 1
>
> Please let us know if these changes would be suitable.

---

> > ### Comment · Reviewer_teX3 · 2025-05-05
> > **Reviewer response**
> >
> > Thank you for addressing my concerns. With the proposed changes I now lean towards acceptance. Going forward I hope more attention variants can be added to strengthen the framework.

---

> > > ### Author Response · Authors · 2025-05-08
> > > **May 8 Update**
> > >
> > > We have updated the abstract to remove the “state-of-the-art” claim and now describe the model as “robust.” We have implemented encoder-compatible versions of Longformer, Linformer, and Perceiver attention methods, and we are currently running a NAS evaluation on the Geneformer task to assess their utility; these results will be included in the revised version, which will be sent out by EOD on May 14th. We have also added code examples to the supplement, performed an ablation experiment to evaluate combinations of positional encodings (finding no additive benefit past the optimal rotary method, thus requiring an edit to our paper), and updated the supplemental figure with a color legend. We additionally removed Big Bird from that same figure to avoid potential confusion. We appreciate the reviewers’ input and look forward to submitting the complete revision shortly.

---

> > > > ### Author Response · Authors · 2025-05-14
> > > > **May 14 Update**
> > > >
> > > > With our new revision submission, we wanted to highlight that we removed the supplementary figure portraying Longformer and Big Bird methods, and thus do not update its color legend as previously advised. We realized as more attention mechanisms were added that doing so for these specific implementations but not the others - including the implemented position encoding strategies - was inconsistent, and largely redundant with the references included. Let us know if you have any thoughts or recommendations on this adjustment.

---

> > > > > ### Comment · Reviewer_teX3 · 2025-05-21
> > > > > **Response to authors**
> > > > >
> > > > > I thank the authors for their response and added discussion and results. I think these changes improve the paper, especially by making the claims more valid and increasing the utility of the package with more attention mechanisms. The added NAS experiment, ablations and code examples are also very welcome. I appreciate the effort put in by the authors in this rebuttal and will recommend acceptance. To ensure the continued use of this package I encourage the authors to keep adding more mechanisms for attention, positional embeddings and other common components.

---

### Decision · Action_Editor_H6qE · 2025-05-22

**Recommendation:** Accept with minor revision

**Comment:**

After a round of corrections by the authors, the reviewers believe the paper should be accepted, and that it will be of interest to the TMLR community. Given the scope and materials in the paper, the authors are expected to release a link to a repository that is accessible and easy to use.

**Audience:**

Practitioners who are interested in using transformers.

**Claims And Evidence:**

The claims made are clear and documented as a description of the developed framework of using transformers (in the form of a software package).

There was a short discussion over email in which the authors requested to make a few minor changes.